

# An inferential study of the phenotype for the chromosome 15q24 microdeletion syndrome: a bootstrap analysis

Antonio Palazón-Bru[1,2], Dolores Ramírez-Prado[3,4], Ernesto Cortés[4], María Soledad Aguilar-Segura[4] and Vicente Francisco Gil-Guillén[1,2]

[1] Department of Clinical Medicine, Miguel Hernández University, San Juan de Alicante, Alicante, Spain
[2] Research Unit, Elda Hospital, Elda, Alicante, Spain
[3] Clinical Analysis Department, Elda Hospital, Elda, Alicante, Spain
[4] Pharmacology, Pediatrics and Organic Chemistry Department, Miguel Hernández University, San Juan de Alicante, Alicante, Spain

## ABSTRACT

In January 2012, a review of the cases of chromosome 15q24 microdeletion syndrome was published. However, this study did not include inferential statistics. The aims of the present study were to update the literature search and calculate confidence intervals for the prevalence of each phenotype using bootstrap methodology. Published case reports of patients with the syndrome that included detailed information about breakpoints and phenotype were sought and 36 were included. Deletions in megabase (Mb) pairs were determined to calculate the size of the interstitial deletion of the phenotypes studied in 2012. To determine confidence intervals for the prevalence of the phenotype and the interstitial loss, we used bootstrap methodology. Using the bootstrap percentiles method, we found wide variability in the prevalence of the different phenotypes (3–100%). The mean interstitial deletion size was 2.72 Mb (95% CI [2.35–3.10 Mb]). In comparison with our work, which expanded the literature search by 45 months, there were differences in the prevalence of 17% of the phenotypes, indicating that more studies are needed to analyze this rare disease.

## INTRODUCTION

Chromosome 15q24 microdeletion syndrome is a rare disease that was first analyzed by *Sharp et al. (2007)*, characterizing the phenotype and genotype of four patients with this syndrome (*Sharp et al., 2007*). Since this publication, further papers have contributed new cases, providing descriptive comparisons of the phenotype and genotype between the new case and those already described (*Klopocki et al., 2008*; *Marshall et al., 2008*; *Andrieux et al., 2009*; *El-Hattab et al., 2009*; *Masurel-Paulet et al., 2009*; *Van Esch et al., 2009*; *El-Hattab et al., 2010*; *McInnes et al., 2010*; *Ng et al., 2011*; *Brun et al., 2012*; *Mefford et al., 2012*; *Narumi et al., 2012*; *Samuelsson, Zagoras & Hafström, 2015*). Five breakpoints (Low-Copy Repeat [LCR] clusters) have been identified where the majority of

Corresponding author
Antonio Palazón-Bru,
antonio.pb23@gmail.com

the microdeletions have occurred: LCR15q24A, LCR15q24B, LCR15q24C, LCR15q24D, and LCR15q24E (*Magoulas & El-Hattab, 2012*).

As a result of these reports, in January 2012 *Magoulas & El-Hattab (2012)* conducted a comprehensive review of the cases reported up to that time (*Magoulas & El-Hattab, 2012*). In this review they descriptively analyzed the phenotypes of the patients with the syndrome, both in detail and in large groups. However, as this work did not include inferential statistics calculating the proportion of children with the different phenotypes with their Confidence Intervals (CI), we conducted a study to update the literature search and to determine the corresponding ranges for each phenotype using bootstrap methodology. The results provide a better understanding of this rare condition.

## MATERIALS AND METHODS

### Study population

The population included all individuals with chromosome 15q24 microdeletion syndrome.

### Study design and participants

This pooled analysis study examined *MEDLINE* and *Google Scholar* databases for reported cases of patients with the syndrome, using the keywords *15q24, deletion* and *microdeletion*. The references in all the cases obtained were reviewed to detect possible studies not found in the databases analyzed, in order to collect all the papers that had been used as references to cases of the syndrome. This search was updated on September 29, 2015.

Starting with the second published case report, the authors compared the characteristics of their patients with the published cases (*Klopocki et al., 2008*; *Marshall et al., 2008*; *Andrieux et al., 2009*; *El-Hattab et al., 2009*; *Masurel-Paulet et al., 2009*; *Van Esch et al., 2009*; *El-Hattab et al., 2010*; *McInnes et al., 2010*; *Ng et al., 2011*; *Brun et al., 2012*; *Mefford et al., 2012*; *Narumi et al., 2012*; *Samuelsson, Zagoras & Hafström, 2015*). We analyzed all the references in these case reports in order to obtain the maximum sample size.

### Variables and measurements

First, we determined which children had each of these phenotypes: male gender, developmental delay, low birth weight/intrauterine growth restriction, short stature, obesity, microcephaly, feeding difficulties, long face, facial asymmetry, high anterior hairline, epicanthal folds, hypertelorism, downslanting palpebral fissures, sparse and broad medial eyebrows, strabismus, nystagmus, broad nasal base, depressed nasal bridge, high nasal bridge, ear abnormalities, palate abnormalities, long smooth philtrum, full lower lip, small mouth, hypospadias, microphallus, cryptorchidism, thumb abnormalities, brachydactyly/short digits, clynodactyly, toe abnormalities, joint laxity, scoliosis/kyphosis, hypotonia, behavior problems, Magnetic Resonance Imaging (MRI) abnormalities, recurrent infections, hernias, congenital heart disease, hearing loss, diaphragmatic hernia, intestinal atresia, imperforate anus, coloboma, dental

problems and myelomeningocele. The phenotypes used followed the previous review (*El-Hattab et al., 2010*). The deleted megabase (Mb) pairs in each patient were determined to calculate the size of the interstitial deletion of chromosome 15q24 (in Mb). Finally, we also analyzed the five breakpoints (*Magoulas & El-Hattab, 2012*).

The source of information was the selected papers, i.e., those containing complete information on phenotype and genotype (Mb).

## Sample size

The sample size was 40 patients, 4 of whom were excluded due to lack of the information required for this study (Mb and phenotype). The final sample consisted of 36 patients.

## Statistical methods

The phenotype was described using absolute frequencies. One thousand bootstrap samples were obtained to determine the relative distribution frequency for the phenotypes and the mean loss size of the interstitial deletion of chromosome 15q24 (in Mb). Using these samples, we calculated a point estimate (median) and a CI (percentiles method) for these parameters (proportions and means). Thus we obtained a measurement of the uncertainty for the relative frequencies and for the mean, since with a small sample size we could not perform an asymptotic approximation of the normal distribution (n = 36). This bootstrap methodology was based on taking 1,000 simple random samples with replacement of the original sample. For each sample we calculated the statistic of interest (in our case proportions/means). At the end of the process we obtained 1,000 values for the statistic. In other words, we had a distribution of the statistic from which we could then calculate its estimation (median) with its CI (P2.5–P97.5). This method is very useful for obtaining the CI for statistics which are difficult to calculate (no closed form or with hypotheses which are not present). The computational cost is the only aspect of this method that is inadequate, but as our sample was small, this was not a problem. This methodology has already been applied with other rare diseases (*Chiano & Yates, 1994*). All analyses were performed with a significance level of 5%. The statistical software used was IBM SPSS Statistics 19.

## RESULTS

Table 1 shows the descriptive and analytical features of all the patients found in the literature search. We highlight the wide variability in the prevalence of the different phenotypes (3–100%). The phenotype with the lowest prevalence was myelomeningocele, which was only present in one case, whereas developmental delay was common in all cases (n = 36). The mean interstitial deletion loss was 2.72 Mb (95% CI [2.35–3.10 Mb]). Regarding the five breakpoints, we obtained the following results (patients who presented the deletion at the breakpoint): A) 19 (0.53, 95% CI [0.36–0.69]); B) 30 (0.83, 95% CI [0.69–0.94]); C) 25 (0.69, 95% CI [0.56–0.86]); D) 11 (0.31, 95% CI [0.17–0.47]); and E) 9 (0.25, 95% CI [0.11–0.39]).

**Table 1 Descriptive and analytical features of the phenotypes of the patients with chromosome 15q24 microdeletion syndrome.**

| Phenotype | Total n = 36 | Proportion (95% CI) | Phenotype | Total n = 36 | Proportion (95% CI) |
|---|---|---|---|---|---|
| Male gender | 27 | 0.75 (0.61–0.89) | Small mouth | 9 | 0.25 (0.11–0.39) |
| Developmental delay | 36 | 1 (1–1) | Hypospadias | 9 | 0.25 (0.11–0.39) |
| Low birth weight/IUGR | 14 | 0.39 (0.22–0.56) | Microphallus[*] | 5 | 0.19 (0.04–0.36) |
| Short stature | 11 | 0.31 (0.17–0.47) | Cryptorchidism | 4 | 0.11 (0.03–0.22) |
| Obesity | 8 | 0.22 (0.08–0.36) | Thumb abnormalities | 8 | 0.22 (0.11–0.36) |
| Microcephaly | 8 | 0.22 (0.08–0.36) | Brachydactyly/short digits | 12 | 0.33 (0.19–0.50) |
| Feeding difficulties | 7 | 0.19 (0.08–0.33) | Clynodactyly | 3 | 0.08 (0.00–0.19) |
| Long face | 9 | 0.25 (0.11–0.42) | Toe abnormalities | 10 | 0.28 (0.14–0.44) |
| Facial asymmetry | 5 | 0.14 (0.03–0.25) | Joint laxity | 14 | 0.39 (0.22–0.56) |
| High anterior hairline | 11 | 0.31 (0.17–0.47) | Scoliosis/Kyphosis | 8 | 0.22 (0.08–0.36) |
| Epicanthal folds | 17 | 0.47 (0.31–0.64) | Hypotonia | 18 | 0.50 (0.33–0.64) |
| Hypertelorism | 8 | 0.22 (0.08–0.36) | Behavior problems | 16 | 0.44 (0.28–0.58) |
| Downslanting palpebral fissures | 15 | 0.42 (0.25–0.58) | MRI abnormalities | 13 | 0.36 (0.19–0.53) |
| Sparse broad medial eyebrows | 15 | 0.42 (0.25–0.58) | Recurrent infections | 11 | 0.31 (0.17–0.44) |
| Strabismus | 10 | 0.28 (0.14–0.44) | Hernias | 5 | 0.14 (0.03–0.28) |
| Nystagmus | 3 | 0.08 (0.00–0.19) | Congenital heart disease | 8 | 0.22 (0.11–0.36) |
| Broad nasal base | 5 | 0.14 (0.03–0.25) | Hearing loss | 10 | 0.28 (0.14–0.42) |
| Depressed nasal bridge | 6 | 0.17 (0.06–0.31) | Diaphragmatic hernia | 2 | 0.06 (0.00–0.14) |
| High nasal bridge | 2 | 0.06 (0.00–0.14) | Intestinal atresia | 2 | 0.06 (0.00–0.14) |
| Ear abnormalities | 21 | 0.58 (0.42–0.75) | Imperforate anus | 2 | 0.06 (0.00–0.14) |
| Palate abnormalities | 8 | 0.22 (0.08–0.36) | Coloboma | 2 | 0.06 (0.00–0.14) |
| Long smooth philtrum | 15 | 0.42 (0.25–0.58) | Dental problems | 3 | 0.08 (0.00–0.19) |
| Full lower lip | 9 | 0.25 (0.14–0.42) | Myelomeningocele | 1 | 0.03 (0.00–0.08) |

**Notes:**
CI, confidence interval; IUGR, intrauterine growth restriction; MRI, magnetic resonance imaging.
[*]Only for male gender.

## DISCUSSION

### Summary

This study updates the descriptive analysis of the different phenotypes conducted in 2012 (*Magoulas & El-Hattab, 2012*), and provides inferential information on a range of values within which the true proportion of each phenotype in the study population can likely be found (95% CI).

### Strengths and limitations of the study

The main strength of this study is that, for the first time, a measurement of the uncertainty of the proportion of each phenotype in the population has been obtained using inferential statistics (bootstrap samples). In addition, the descriptive data provided by *Magoulas & El-Hattab (2012)* have also been updated.

Concerning the limitations of this study, we have to assume a possible selection bias, since it is possible that not all incident cases of this syndrome have been published or

that in our search strategy we may not have found all recently published cases, as we did not include the DECIPHER and ECARUCA databases. To minimize this bias as much as possible, we rigorously reviewed all the references for each of the articles. In terms of reporting bias, it is possible that not all the features have been specified in each of the cases. For example, MRI abnormalities: not all reported patients would have had an MRI scan done. In those cases the statistics for MRI abnormalities would be difficult to derive. Regarding obesity, the other authors used the Z-score for its assessment. In other words, the researchers took into account the age and gender of the patients. Other features are largely descriptive (thumb abnormalities, hypertelorism and brachydatyly) and thus subject to single clinical observer bias. Finally, the information was obtained through the publications instead of by contacting the authors. For obvious reasons, this information bias cannot be minimized. If we were analyzing associations between phenotypes and genotypes, this bias could give us incorrect information about the associations. However, we only obtained descriptive statistics with their confidence interval using the original sample through bootstrap methodology.

## Comparison with the existing literature

To compare our results with those found by the 2012 study (*Magoulas & El-Hattab, 2012*), we determined whether the point estimate given by the previous study was within the CI we constructed for each phenotype. Such that if the estimate of the other study fell within that range, we had no evidence indicating that the proportion was different from that of our study (*Magoulas & El-Hattab, 2012*). However, if this estimate fell outside our CI, there was a difference compared with our study.

Overall we found agreement between both studies, as the previous estimate in nearly 83% of the phenotypes (38/46) was within our CI. The eight phenotypes that showed discrepancies with the previous study (*Magoulas & El-Hattab, 2012*) were facial asymmetry, high anterior hairline, hypertelorism, broad nasal base, hypospadias, microphallus, thumb abnormalities, and clynodactyly, all of which were approximately twice as prevalent in the earlier study (*Magoulas & El-Hattab, 2012*) than in our study. To assess the reasons for these discrepancies we need further studies on this syndrome, because they cannot be determined using our study design.

## Implications for research and/or practice

The comparative analysis between our results and those of the review published in 2012 (*Magoulas & El-Hattab, 2012*) revealed differences in 17% of the phenotypes. Given that the previous article was accepted for publication in January 2012 (*Magoulas & El-Hattab, 2012*), and that our work expanded the literature analysis by approximately 45 months, it appears that the prevalence of the phenotypes in these patients is not fully known, which suggests the need for future studies in this rare syndrome. When there are more published cases of this rare syndrome, it would be very interesting to determine associations between the phenotypes and the size of the deletion. These associations were not studied in our paper because with 36 patients we would not have been able to obtain valid conclusions. Finally, it would be interesting to measure some phenotypes with facial

recognition software (facial asymmetry, high anterior hairline, hypertelorism and broad nasal base).

Once we have this new knowledge, we can devise strategies to improve our decisions regarding these patients. For instance, we could attempt to determine where the breakpoint is using only the information about the phenotype.

## CONCLUSIONS

The prevalence of the different phenotypes in patients with chromosome 15q24 microdeletion syndrome, published in early 2012 (*Magoulas & El-Hattab, 2012*), has been updated. The CI were calculated based on bootstrap samples for these proportions, providing a measurement of the uncertainty in the study population. Furthermore, differences between our study and the 2012 study were found in approximately one in six phenotypes, which suggests more studies are needed to analyze the characteristics of this rare syndrome.

## ACKNOWLEDGEMENTS

We thank Maria Repice and Ian Johnstone for their help in preparing the English version of this work.

### Funding

The authors received no funding for this work.

### Competing Interests

Antonio Palazón-Bru is an Academic Editor for PeerJ.

### Author Contributions

- Antonio Palazón-Bru conceived and designed the experiments, performed the experiments, analyzed the data, wrote the paper, prepared figures and/or tables, reviewed drafts of the paper.
- Dolores Ramírez-Prado conceived and designed the experiments, performed the experiments, wrote the paper, prepared figures and/or tables, reviewed drafts of the paper.
- Ernesto Cortés conceived and designed the experiments, reviewed drafts of the paper.
- María Soledad Aguilar-Segura conceived and designed the experiments, reviewed drafts of the paper.
- Vicente Francisco Gil-Guillén conceived and designed the experiments, reviewed drafts of the paper.

### Data Deposition

We have provided the raw data as a Supplementary Dataset.

## Supplemental Information

Supplemental information for this article can be found online at http://dx.doi.org/10.7717/peerj.1641#supplemental-information.

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
