# Peer review of "An inferential study of the phenotype for the chromosome 15q24 microdeletion syndrome: a bootstrap analysis"

_PeerJ, doi:10.7717/peerj.1641_

## Round 0.1 · original submission · Major Revisions

Please revise your manuscript with more justification and description of the boot strap method. There are concerns raised by three reviewers of the methodology and its appropriateness.

In addition to the three reviews please also respond to.

Experimental design
1. The experiments have been designed to provide statistical measure of different phenotypes associated with 15q24 deletions. The authors correctly identify selection and reporting biases as important limitations of their study.

2. My question is how can authors be sure if 'not reported' is same as 'not checked for' or 'absent'? For example - MRI abnormalities - not all reported patients would have had MRI scan done. In that case the statistics for MRI abnormalities may be difficult to derive.

3. Some phenotypes may only emerge beyond a certain age - e.g. - obesity. Is it fair to give equal weight to this phenotype in all patients without considering their age.

4. Some phenotypes may be sex specific - e.g. microphallus. It is not correct to use the same 'n=36' when calculating the proportion for this phenotype.

5. Furthermore a number of phenotypes can be relatively subjective. For example - hypertelorism and brachydatyly.

6. Please clarify what is meant by the sentence - "One thousand bootstrap samples were obtained."

Validity of the findings
1. Some of the above mentioned limitations may perhaps be ignored in a descriptive study but are harder to avoid when developing an accurate statistical model. It is, therefore, difficult to be confident of the validity of the findings, unless adequately addressed by the authors.

2. Comparison with the existing literature - Eight features showed discrepancy in comparison with a previous review on this condition. The authors list these features but do not explain or hypothesize why this might be so.

3. Authors mention that "the prevalence of the phenotypes in these patients is not stable". This statement may be open to misinterpretation. Did you want to say that the frequency of different phenotypes is not fully known?

4. The experimental design included documentation of the size of the deletion. However, this is not reflected in the results and not discussed subsequently. Is their any correlation between the phenotypes and size of the deletion?

·

Basic reporting

No major concerns. Well structured. Contains some typographical errors.

Experimental design

The authors have looked at frequency of specific characteristics in the 15q24 syndrome. they have only looked at data from published articles whereas much of the data on microdeletion syndromes nowadays is held in databases such as DECIPHER and ECARUCA. They could have improved their sample size by contacting those who contributed cases to these databases and seeking further information.

They have looked at the overall size of deletions but the actual position of the deletion and the gene content is much more important than the size.

Validity of the findings

I am not familiar with bootstarp analysis and connot comment on the statistical validity.
I have not been convinced by this article that Bootstrap analysis is an appropriate method of analysing the very variable features seen in microdeletion syndromes. Data was obtained solely from literature rather than by contacting authors to get deeper phenotype information and this is a significant limitation.

Additional comments

Diddecting the clinical phenotypes of microdeletion syndromes can be useful - there are many good examples of how genes for specific characteristics have been identified this way but this type of analysis needs careful consideration of the exact location of deletions, not just the size and very good clinical phenotyping which is not usually available in the literature.

·

Basic reporting

No comments

Experimental design

No comments

Validity of the findings

No comments

Additional comments

The paper is well-written and updates our knowledge about this rare syndrome. It uses appropriate statistical methods to present the prevalence of different phenotypic features of this syndrome. The paper carries a clear value and suggests a better way of evaluating the clinical features associated with such rare syndromes.

·

Basic reporting

This is a new approach in assigning a range of phenotypic variability to a mutation. There are some minor errors in English and in the descriptive medical terms, for example clynodactily should be change to clinodactyly, brachydactily to brachydactyly,

Experimental design

The authors should explain in detail what is bootstrapping and why they chose this inferential statistics method to analyse this data. Has this been used before in analysing phenotypic data connected with a specific mutation? What are the advantages of this statistical method in analysing phenotypic data?
Also, in the 'Study population' they mention that 'All individuals with a 15q24 microdeletion were included. I presume they mean all reported individuals in the medical literature so far.
Have the authors thought of taking into consideration other reported populations of the 15q24 microdeletion like, for example, the one included in the DECIPHER database?

Validity of the findings

The authors report discordant results with previous studies (a lower prevalence) for a number of phenotypic data. A few of these can be measurable with facial recognition softwares (facial asymmetry, high anterior hairline, hypertelorism, broad nasal base) and I think the authors should mention this as an implication for future research. Others are largely descriptive (thumb abnormalities) and thus subject to bias of the single clinical observer and this should be mentioned in the limitations.
Finally a comment should be made on what are the implications of these findings for patient management and if this research modified our knowledge around the 15q24 microdeletion patients in a medically/clinically actionable manner.

Reviewer 4 ·

Basic reporting

1. The paper describes bootstrapping in cases of 15q24 deletions. This is a novel approach and the paper will be of interest to geneticists and clinicians.

2. A better background with description of what is "bootstrapping", its general advantages and limitations would help the readability of the paper.

3. Additionally, it would be helpful to cite published examples where bootstrapping has been successfully employed in rare Mendelian disorders to reassure that 'this approach has been shown to work'.

Experimental design

1. The experiments have been designed to provide statistical measure of different phenotypes associated with 15q24 deletions. The authors correctly identify selection and reporting biases as important limitations of their study.

2. My question is how can authors be sure if 'not reported' is same as 'not checked for' or 'absent'? For example - MRI abnormalities - not all reported patients would have had MRI scan done. In that case the statistics for MRI abnormalities may be difficult to derive.

3. Some phenotypes may only emerge beyond a certain age - e.g. - obesity. Is it fair to give equal weight to this phenotype in all patients without considering their age.

4. Some phenotypes may be sex specific - e.g. microphallus. It is not correct to use the same 'n=36' when calculating the proportion for this phenotype.

5. Furthermore a number of phenotypes can be relatively subjective. For example - hypertelorism and brachydatyly.

6. Please clarify what is meant by the sentence - "One thousand bootstrap samples were obtained."

Validity of the findings

1. Some of the above mentioned limitations may perhaps be ignored in a descriptive study but are harder to avoid when developing an accurate statistical model. It is, therefore, difficult to be confident of the validity of the findings, unless adequately addressed by the authors.

2. Comparison with the existing literature - Eight features showed discrepancy in comparison with a previous review on this condition. The authors list these features but do not explain or hypothesize why this might be so.

3. Authors mention that "the prevalence of the phenotypes in these patients is not stable". This statement may be open to misinterpretation. Did you want to say that the frequency of different phenotypes is not fully known?

4. The experimental design included documentation of the size of the deletion. However, this is not reflected in the results and not discussed subsequently. Is their any correlation between the phenotypes and size of the deletion?

Additional comments

No comments

---

## Round 0.2 · accepted · Accept

You have responded appropriately to the reviewer comments and explained the boot strap method